# Generalization, robustness and adaptability of Progressive Neural Collapse

## Abstract

Neural networks exhibit the neural collapse phenomenon in multi-class classification tasks, where last-layer features and linear classifier weights converge into a symmetric geometric structure. However, most prior studies have primarily focused on last-layer feature representations or have examined intermediate features using limited, simple architectures and datasets. The mechanisms by which deep neural networks separate data according to class membership across all layers in more complex and realistic scenarios, and how this separation evolves under distribution shifts, remain unclear. In this work, we extend the study of neural collapse to a broader range of architectures and datasets, investigating its progression throughout the network and its implications for generalization, robustness, and domain adaptability. Our findings reveal that well-trained neural networks progressively enhance neural collapse across layers, though a distinct transition phase occurs where this improvement plateaus after the initial layers and is followed by a renewed continuous improvement in the very last layers, with additional layers contributing minimal generalization benefits. Moreover, we observe that this progressive neural collapse pattern remains robust against noisy data, whether the noise occurs in inputs or labels, and that the degree of intermediate separation serves as an effective indicator of noise levels. Additionally, for the learned networks, comparing neural collapse evaluated on noisy data and clean data reveals insights into feature learning and memorization, with the latter primarily occurring in the very last layers. This finding aligns with the neural collapse pattern observed with clean training data. Finally, we show that when a shift occurs between source and target domains, intermediate neural collapse is closely related to downstream target performance.

## 1 Introduction

Deep learning has become the de facto choice for a wide range of machine learning applications, including image recognition (He et al., 2016; Radford et al., 2021), language modeling (Vaswani, 2017; Devlin et al., 2018), and scientific computing (Silver et al., 2016; Fawzi et al., 2022). These models are increasingly applied in diverse real-world scenarios, such as handling corrupted inputs, noisy labels, and domain adaptation tasks. However, despite their widespread success, the underlying reasons for the remarkable generalization abilities of deep networks remain poorly understood. Much of their success has been attributed to the ability to learn hierarchical representations, which enables deep learning models to capture complex patterns across different layers (Bengio et al., 2013). Yet, the mechanisms behind their robustness and adaptability in challenging environments, such as those involving input corruption or shifts in data distributions, are still not fully explained. In this paper we are motivated by the following question: *how to characterize hierarchical representations, and how robust and adaptable are they in the presence of labeling noise, corrupted inputs, and domain shifts?*

Papyan et al. (2020) empirically identified an intriguing phenomenon termed *Neural Collapse* ($\mathcal{NC}$) for the balanced multi-class classification tasks. During the terminal phase of training, once the training error reaches zero, both the last-layer features and the final linear classifier converge to a highly symmetric and structured geometric configuration. Specifically, the last-layer features collapse to their corresponding class means ($\mathcal{NC}_1$), and the class-mean features themselves are maximally distant, forming a simplex equiangular tight frame (ETF) structure ($\mathcal{NC}_2$). Simultaneously,

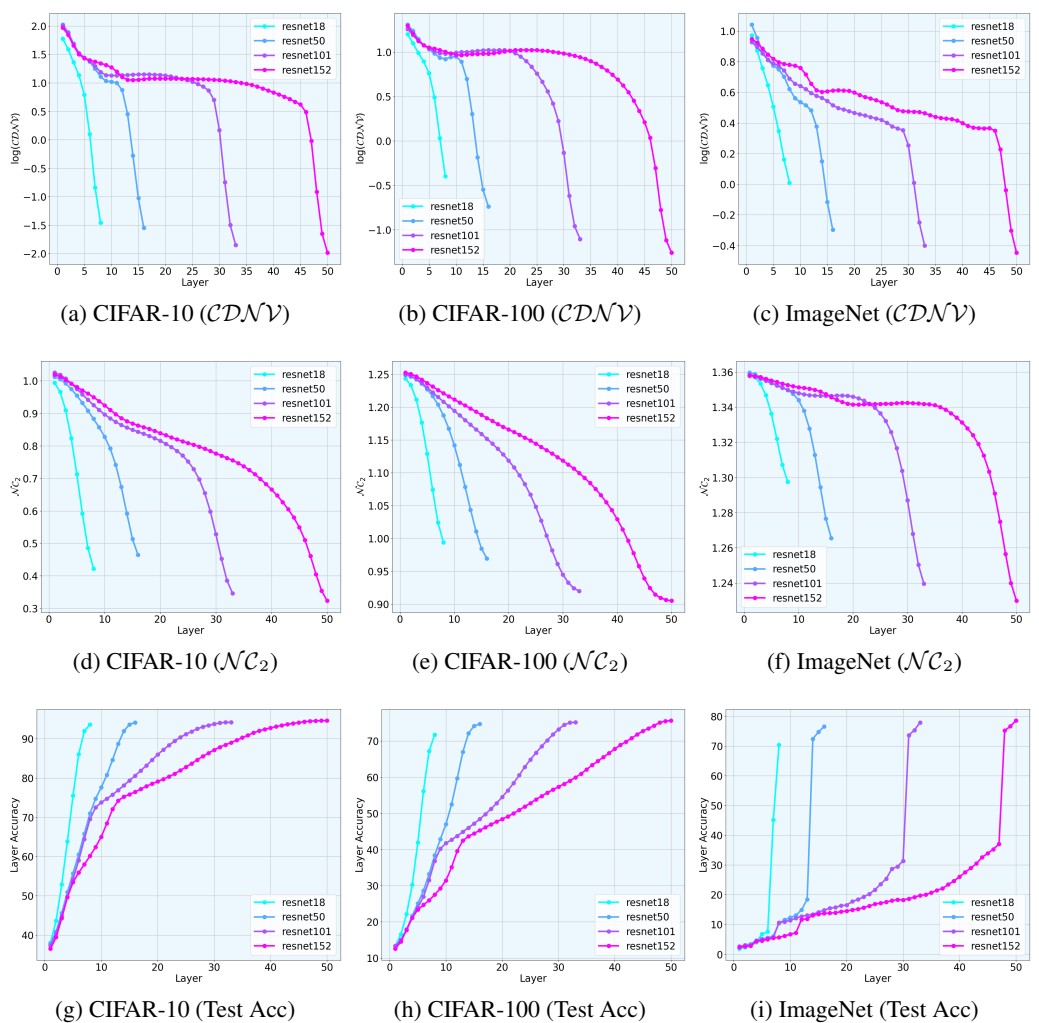

Figure 1: **The evolution of intermediate feature separation across layers for ResNet based models on different dataset.** The graphs depict the layer-wise progression of within-class variability(top row),between-class separation (middle row) and layer-wise linear-probing accuracy (bottom row) on CIFAR-10, CIFAR-100 and ImageNet datasets for different ResNet architectures.

the classifier weights align perfectly with the centered class-mean features, up to a scaling factor ($\mathcal{NC}_3$). Consequently, this geometric structure leads the classifier to make predictions by selecting the class with the nearest train class mean ($\mathcal{NC}_4$).

Neural Collapse offers a mathematically elegant characterization of the learned representations in the penultimate layer of deep learning-based classification models, independent of network architecture and dataset. While the research in this field has enhanced the understanding of how deep neural networks functions from different perspectives, most existing theoretical and empirical work focuses on last-layer features or examines intermediate features using relatively small network architectures, such as MLP, VGG, and shallow ResNet, and on simpler datasets like MNIST, Fashion-MNIST, and CIFAR-10. For example, He & Su (2023) suggests that intermediate layers in deep networks enhance the within-class variability at a constant geometric rate. Nonetheless, this phenomenon has mostly been observed with simple architectures and datasets, leaving open questions about its validity in more complex, real-world scenarios.

**Contributions.** In this work, we conduct an extensive empirical investigation across a diverse set of computer vision datasets, focusing on the intermediate representations of several contemporary neural networks in real-world scenarios. Our contributions can be summarized as follows:

- **Progressive intermediate neural collapse with a phase transition as depth increases.** Contrary to the geometric rate of within-class collapse reported by He & Su (2023), our findings show that well-trained neural networks indeed progressively enhance neural collapse across layers, although this geometric rate is not consistently observed in more complex settings. To better understand the interplay between network depth and dataset complexity on layer-wise neural collapse, we analyze how intermediate neural collapse evolves as network depth increases across various datasets. We observe a distinct transition phase of the within-class collapse as illustrated in Figure 1: for shallower networks relative to the dataset complexity, with-class collapse improves steadily across layers. However, as network depth increases, a transition phase emerges where the initial improvement reaches a plateau, and additional layers provide minimal benefit. Interestingly, after this plateau, the final layers show a renewed continuous improvement in within-class collapse.

- **Marginal gains in generalization with more layers after transition phase occurs.** Moving forward, we raise the question of whether this transitional phenomenon observed in intermediate features is connected to generalization performance. Notably, our findings indicate that when this phenomenon occurs, increasing model depth leads to only marginal gains in generalization performance. In contrast to conventional approaches, which require a separate validation set to determine the smallest depth for maximizing generalization, our results suggest that this transitional behavior could serve as an intrinsic indicator for identifying the most efficient depth, beyond which additional depth yields diminishing returns.

- **Intermediate neural collapse under distribution shift.** We argue that the presence of a plateau region in the middle layers, followed by an accelerated decay in compression and separation in the final layers, benefits the generalization and transferability of DNNs. To support this argument, we investigate intermediate neural collapse across three practical scenarios: label noise, corrupted input, and domain shift. We observe that the intermediate features of the training data continue to exhibit progressive neural collapse, with patterns remaining consistently similar to those seen in clean data, although the degree of collapse varies depending on the noise level or domain shifts. For training with noisy data (label noise or corrupted input), we define the *memorization ratio* for each layer as the ratio of neural collapse evaluated on clean data to that on noisy data. Notably, regardless of network size, the memorization ratio remains below 1 for all layers except the final few ones, indicating that memorization primarily occurs in the last layers, while preceding layers learn meaningful representations. For domain shift, our findings reveal that intermediate features exhibiting greater neural collapse on downstream target data tend to demonstrate better adaptability and yield higher linear-probing accuracy.

## 2 RELATED WORKS

**Last-layer neural collapse.** The $\mathcal{NC}$ phenomenon was first discovered in Papyan et al. (2020).Under the assumption of the *unconstrained feature model*(UFM) Mixon et al. (2022); Fang et al. (2021), which treats last-layer features as free optimization variables, a series of theoretical studies have validated the existence of the $\mathcal{NC}$ phenomenon. For example, studies such as Mixon et al. (2022); Han et al. (2021); Zhu et al. (2021); Zhou et al. (2022a); Lu & Steinerberger (2022); Fang et al. (2021); Ji et al. (2021); Tirer & Bruna (2022); Yaras et al. (2022); Zhou et al. (2022b); Fisher et al. (2024) demonstrated that the global minimizers satisfy the $\mathcal{NC}$ properties for a family of loss functions, including cross-entropy loss, mean-square-error loss, and label-smoothing loss, among others, when the last-layer feature dimension is not smaller than the number of classes. Moreover, when the number of classes is sufficiently large, Jiang et al. (2023); Gao et al. (2023) proved that the last-layer features satisfy generalized $\mathcal{NC}$ properties. Beyond UFM, Tirer & Bruna (2022) and Dang et al. (2023) characterize the global optimality of a two-layers models and multi linear layer models, respectively. Súkeník et al. (2024) extended UFM to arbitrary non-linear layers and proved that $\mathcal{NC}$ emerges after a certain layer for binary classification. These work not only contribute to a new understanding of the working of DNNs but also has also inspired the development of novel techniques across various applications, such as imbalanced learning Xie et al. (2023); Liu et al. (2023), trans-

fer learning Galanti et al. (2022b); Li et al. (2022); Xie et al. (2023); Galanti et al. (2021), and adversarial robustness Su et al. (2023).

**Intermediate neural collapse.** While $\mathcal{NC}$ was initially introduced to describe the configurations of last-layer features, recent studies have extended its investigation to intermediate representations. Tirer et al. (2023) provided a theoretical analysis showing that the within-class variability ($\mathcal{NC}_1$) metric decreases monotonically along the gradient flow across layers when the network is trained using cascade learning, where a new layer is added on top of the pre-trained network at each step. However, this theoretical result does not fully align with the more common practice of training models in an end-to-end manner. Apart from the theoretical results, some empirical studies Hui et al. (2022); Rangamani et al. (2023); He & Su (2023) suggest that the within-class variability ($\mathcal{NC}_1$) of intermediate features decreases monotonically as layers progress deeper into the network. Similarly, research by Ben-Shaul & Dekel (2022); Galanti et al. (2022a) demonstrates that intermediate layers gradually improve the nearest class-center accuracy ($\mathcal{NC}_4$). Rather than focusing on individual $\mathcal{NC}$ properties, recent works Rangamani et al. (2023); Parker et al. (2023); Wang et al. (2024) have extended the analysis to encompass all $\mathcal{NC}$ properties across intermediate layers. However, all of these studies investigate intermediate $\mathcal{NC}$ using relatively small network architectures, such as MLP, VGG, and shallow ResNet, and simpler datasets like MNIST, Fashion-MNIST, and CIFAR-10, which may limit the generalizability of their findings to more complex architectures and datasets.

## 3 THE PROBLEM SETUP

**Notations and Organization.** Throughout the paper, we use bold lowercase and upper letters, such as $\boldsymbol{a}$ and $\boldsymbol{A}$, to denote vectors and matrices, respectively. Not-bold letters are reserved for scalars. The symbols $\boldsymbol{I}_K$ and $\boldsymbol{1}_K$ respectively represent the identity matrix and the all-ones vector with an appropriate size of $K$, where $K$ is some positive integer. We use $[K] := \{1; 2; \cdots; K\}$ to denote the set of all indices up to $K$. For any matrix $\boldsymbol{A} \in \mathbb{R}^{n_1 \times n_2}$, we write $A = [\boldsymbol{a}_1 \quad \cdots \quad \boldsymbol{a}_{n_2}]$, so that $\boldsymbol{a}_i$ ($i \in [n_2]$) denotes the $i$-th column vector of $\boldsymbol{A}$.

The remainder of this section is organized as follows. In section 3.1, we first review deep neural networks. Subsequently, in section 3.2, we review domain adaptation and introduce three practical scenarios that are the focus of this study. Finally, we introduce the neural collapse phenomenon and present the metrics used to measure intermediate $\mathcal{NC}$ in section 3.3.

### 3.1 BASICS OF DEEP NEURAL NETWORKS

Consider a multi-class classification problem with $K$ classes, where each class has $n$ samples $\{\boldsymbol{x}_{k,i}, \boldsymbol{y}_t\}$ i.i.d. sampled from some unknown distributions $\mathcal{P}$. The label of the $i$-th sample $\boldsymbol{x}_{k,i} \in \mathbb{R}^D$ in the $k$-th class is represented by a one-hot vector $\boldsymbol{y}_k \in \mathbb{R}^K$ with unity only in $k$-th entry ($1 \leq k \leq K$). To learn the underlying mapping from the input instance $\boldsymbol{x}_{k,i}$ to their corresponding label $\boldsymbol{y}_k$, deep neural networks stand out among a family of parameterized functions due to their outstanding performance. A typical deep neural network $\Phi_{\boldsymbol{\Theta}}(\boldsymbol{x}_{k,i})$ comprises a encoder network $\phi_{\theta_L}(\boldsymbol{x}_{k,i})$ with $L$ non-linear layers arranged in a layer-wise fashion, followed by a linear classifier $\{\boldsymbol{W}_{L+1}, \boldsymbol{h}_{L+1}\}$, which can be expressed as:

$$\Phi_{\boldsymbol{\Theta}}(\boldsymbol{x}_{k,i}) = \boldsymbol{W}_{L+1} \cdot \phi_{\theta_L}(\boldsymbol{x}_{k,i}) + \boldsymbol{b}_{L+1}; \tag{1}$$

$$\text{and} \quad \phi_{\theta_l}(\boldsymbol{x}_{k,i}) = \sigma\left(\boldsymbol{W}_l \cdot \phi_{\theta_{l-1}}(\boldsymbol{x}_{k,i}) + \boldsymbol{b}_l\right), \quad \text{where} \quad 1 \leq l \leq L; \tag{2}$$

$$\text{and} \quad \phi_{\theta_0}(\boldsymbol{x}_{k,i}) = \boldsymbol{x}_{k,i}, \tag{3}$$

where $\boldsymbol{W}_{L+1}$ and $\boldsymbol{b}_{L+1}$ represents the weight and bias terms of last-layer linear classifier, respectively. For a $L$-layer encoder network $\phi_{\theta_L}(\boldsymbol{x}_{k,i})$, each layer (e.g., the $l$-th layer where $1 \leq l \leq L$) is composed of an affine transformation $\{\boldsymbol{W}_l, \boldsymbol{b}_l\}$, followed by a nonlinear activation $\sigma(\cdot)$ and some normalization functions (e.g., BatchNorm), to extract hierarchical expressive features $\{\phi_{\theta_l}(\boldsymbol{x}_{k,i})\}_{l=1}^L$ from the underlying input instance $\boldsymbol{x}_{k,i}$. For simplicity, we use $\boldsymbol{\Theta}$ to denote all parameters $\{\boldsymbol{W}_l, \boldsymbol{b}_l\}_{l=1}^{L+1}$ of the entire networks and $\theta_l$ to denote the entire parameters of the first $l$-th layers in the encoder networks for $\forall l \in [L]$, where $\theta_L$ represents the all parameters $\{\boldsymbol{W}_l, \boldsymbol{b}_l\}_{l=1}^L$ of the encoder networks. To learn an effective deep classifier, the network parameters, the network parameters $\boldsymbol{\Theta}$ are optimized by minimizing the following empirical risk over the entire $N = nK$

training samples:

$$\boldsymbol{\Theta} := \{\boldsymbol{W}_l, \boldsymbol{b}_l\}_{l=1}^{L+1} := \{\theta_L, \boldsymbol{W}_{L+1}, \boldsymbol{b}_{L+1}\} := \arg\min_{\boldsymbol{\Theta}} \frac{1}{nK} \sum_{t=1}^{K} \sum_{i=1}^{n} \mathcal{L}\left(\boldsymbol{\Phi}_{\boldsymbol{\Theta}}(\boldsymbol{x}_{k,i}), \boldsymbol{y}_k\right),$$

where $\mathcal{L}(\boldsymbol{\Phi}_{\boldsymbol{\Theta}}(\boldsymbol{x}_{k,i}), \boldsymbol{y}) : \mathbb{R}^K \times \mathbb{R}^K \to \mathbb{R}^+$ is a specified loss function which appropriately measure the discrepancy between the prediction $\boldsymbol{\Phi}_{\boldsymbol{\Theta}}(\boldsymbol{x}_{k,i})$ and its corresponding label $\boldsymbol{y}_k$.

## 3.2 BASIC OF DOMAIN ADAPTATION

However, when acquiring label for the target label is difficult and the classification problem is complex, it becomes challenging to learn an effective deep classifier accurately fitting the intricate inherent mapping. To facilitate the development of an effective classifier, a practical solution is to employ domain adaptation, where general feature encoder networks $\phi_{\theta_L}(\cdot)$ are learned through auxiliary relevant *source* tasks and applied on *target* tasks. The underlying rationale is that the source task, with more available labelled data, helps the encoder network to learn more expressive feature representations. Subsequently, the linear classifiers are trained to solve hopefully simpler target classification problems based on the features from the pre-trained encoder network. Since the source task and target task differ only in data source, we will use superscripts $S$ and $T$ to distinguish them for clarity. To conceptualize this problem, given an auxiliary $K$-class classification problem with $n^S$ samples $\left\{\boldsymbol{x}_{k,i}^S, \boldsymbol{y}_k^S\right\}$ ($k \in [K]$ and $i \in \left[n^S\right]$) in each class i.i.d. sampled from an unknown source distribution $\mathcal{P}^S$, the model is initially trained via minimizing a specified loss function $\mathcal{L}^S\left(\boldsymbol{\Phi}_{\boldsymbol{\Theta}}(\boldsymbol{x}_{k,i}^S), \boldsymbol{y}_k^S\right)$ over this source task as follows:

$$\boldsymbol{\Theta}^S := \left\{\theta_L^S, \boldsymbol{W}_{L+1}^S, \boldsymbol{b}_{L+1}^S\right\} := \arg\min_{\boldsymbol{\Theta}} \frac{1}{n^S K} \sum_{k=1}^{K} \sum_{i=1}^{n^S} \mathcal{L}^S\left(\boldsymbol{\Phi}_{\boldsymbol{\Theta}}(\boldsymbol{x}_{k,i}^S), \boldsymbol{y}_k^S\right). \tag{4}$$

After pre-training, the model can then be effectively adapted to a wide range of downstream target tasks by either fine-tuning the entire network parameters or by linear probing a series of linear classifiers that leverage the hierarchical features from the pre-trained encoder networks. In this work, we focus on the layer-wise linear probing method. On one hand, linear probing reflects the quality of deep representations after the neural network is sufficiently trained. On the other hand, linear probing not only becomes more computationally efficient as the model size explosively grows, but also demonstrates competitive or even superior performance compared to full model fine-tuning in many practical tasks Xie et al. (2022); Galanti et al. (2021); Yang et al. (2023); Tian et al. (2020); Kumar et al. (2022). Therefore, for the target $K$ classification task with $n^S$ samples $\left\{\boldsymbol{x}_{k,i}^T, \boldsymbol{y}_k^T\right\}$ ($k \in [K]$ and $i \in \left[n^T\right]$) in each class i.i.d. sampled from an unknown source distribution $\mathcal{P}^T$, the linear classifiers of each layer can be optimized via minimizing the loss function $\mathcal{L}^T$ between the $i$-th layer prediction $\bar{\boldsymbol{W}}_l \cdot \phi_{\theta_l^S}(\boldsymbol{x}_{k,i}) + \bar{\boldsymbol{b}}_l$ and its corresponding label $\boldsymbol{y}_k^T$ as follows:

$$\left\{\bar{\boldsymbol{W}}_l^S, \bar{\boldsymbol{b}}_l^S\right\} := \arg\min_{\left\{\bar{\boldsymbol{W}}_l, \bar{\boldsymbol{b}}_l\right\}} \frac{1}{n^T K^T} \sum_{k=1}^{K^T} \sum_{i=1}^{n^T} \mathcal{L}^T\left(\bar{\boldsymbol{W}}_l \cdot \phi_{\theta_l^S}(\boldsymbol{x}_{k,i}) + \bar{\boldsymbol{b}}_l, \boldsymbol{y}_k^T\right), \tag{5}$$

where $\left\{\bar{\boldsymbol{W}}_l, \bar{\boldsymbol{b}}_l\right\}$ denotes the parameters of the linear classifier which performs linear-probing based on the features $\phi_{\theta_l^S}(\boldsymbol{x}_{k,i})$ from the $l$-th layer. Note that while the features $\phi_{\theta_l^S}(\boldsymbol{x}_{k,i})$ is obtained from the input instance $\boldsymbol{x}_{k,i}^T$ in the target dataset, the training of the parameters $\theta_l^S$ in the encoder network is conducted on the source dataset, which is fully agnostic of the target task. Therefore, to simplify the notation we will drop the $S$ superscription in $\theta_l^S$ whenever this does not cause any confusion.

Denote by $\mathcal{P}^S = \mathcal{P}_{\mathcal{X}}^S \times \mathcal{P}_{\mathcal{Y}}^S$ (and $\mathcal{P}^T = \mathcal{P}_{\mathcal{X}}^T \times \mathcal{P}_{\mathcal{Y}}^T$) the joint distribution of the source (and target) over the input space $\mathcal{X}$ and label space $\mathcal{Y}$. Since the distribution shift between the source task and target task varies across different scenarios, based on the types of differences, we examine three practical scenarios under the recent discovery of *Neural Collapse* in this study:

(1) **Label noise** ($\mathcal{P}_{\mathcal{X}}^S = \mathcal{P}_{\mathcal{X}}^T$ but $\mathcal{P}^S(\mathcal{Y}|\boldsymbol{x}) \neq \mathcal{P}^T(\mathcal{Y}|\boldsymbol{x})$ for some $\boldsymbol{x} \in \mathcal{X}$): Variations in human judgment and the labor-intensive nature of labeling can result in inaccuracies in the source dataset's labels. In this scenario, we hypothesize that while the source and target distributions are largely similar, a small portion of the data might be mislabeled.

(2) **Corrupted Input** ($\mathcal{P}_{\mathcal{X}}^T \neq \mathcal{P}_{\mathcal{X}}^T$ but $\mathcal{P}^S(\mathcal{Y}|\mathcal{X}) = \mathcal{P}^T(\mathcal{Y}|\mathcal{X})$): Due to unavoidable equipment noise and environmental changes during image acquisition, the target distribution diverges from the source, with samples captured under varying noise, blur, and lighting conditions. In this scenario, we assume the semantics of the source and target distributions remain largely similar.

(3) **Domain Shift** ($\mathcal{P}_{\mathcal{X}}^S \neq \mathcal{P}_{\mathcal{X}}^T$ but $\mathcal{P}^S(\mathcal{Y}|\mathcal{X}) = \mathcal{P}^T(\mathcal{Y}|\mathcal{X})$): The reuse of pretrained models as starting points has demonstrated widespread success in fields such as computer vision, natural language processing, and reinforcement learning Zhuang et al. (2020); Devlin et al. (2018); Zhu et al. (2023), even when significant differences exist between the source and target domains, such as using CT images for MRI or virtual video games for real-world simulations.

### 3.3 NEURAL COLLAPSE

Neural Collapse ($\mathcal{NC}$) is an universal phenomenon observed in the last-layer features and the linear classifier of deep neural networks trained on classification problems. During the terminal phase of training (TPT), when the training reaches perfect accuracy, several appealing properties emerge, including the collapsed within-class feature variability and the maximal equiangular separation among class centers of features from different classes. For notation simplification, we drop the subscription $S$ and $T$ without consideration of data sources, and simplify the notation of $i$-th layer features $\phi_{\theta_l}(\boldsymbol{x}_{k,i})$ as $\boldsymbol{h}_{l,k,i}$ for $\forall l \in [L]$, $k \in [K]$ and $i \in [n]$. Additionally, we denote the global mean $\overline{\overline{\boldsymbol{h}}}_l$ and $k$-th class mean $\overline{\boldsymbol{h}}_{l,k}$ of $i$-th layer features as $\overline{\overline{\boldsymbol{h}}}_l = \frac{1}{nK} \sum_{k=1}^{K} \sum_{i=1}^{n} \boldsymbol{h}_{l,k,i}$ and $\overline{\boldsymbol{h}}_{l,k} = \frac{1}{n} \sum_{i=1}^{n} \boldsymbol{h}_{l,k,i}$. Therefore, these two $\mathcal{NC}$ properties of last-layer (e.g. $l = L$) features can be expressed as follows:

- **Within-class variability collapse.** In each class, the last-layer features converges to their corresponding class-mean centers with zero variability,

$$\sigma_{L,k} = \frac{1}{n} \sum_{i=1}^{n} \left\| \boldsymbol{h}_{L,k,i} - \overline{\mathbf{h}}_{L,k} \right\|_2^2 \to 0, \quad \forall k \in [K], \ i \in [n]. \tag{6}$$

Inspired by foundational works (Fisher, 1936; Rao, 1948), $\mathcal{NC}_1$ was originally quantified for the last-layer features using an inverse *signal-to-noise ratio* (SNR), which depends on the ratio of within-class variability to between-class variability. To measure the within-class variability of intermediate features, we employ the class-distance normalized variance (CDNV) proposed by Galanti et al. (2021) and extend it to the intermediate features:

$$\text{CDNV}_{l,k,k'} := \frac{\sigma_{l,k}^2 + \sigma_{l,k'}^2}{2 \left\| \overline{\boldsymbol{h}}_{l,k} - \overline{\boldsymbol{h}}_{l,k'} \right\|_2^2}, \quad \forall k \neq k', l \in [L]. \tag{7}$$

These pair-wise measures constitute the off-diagonal entries of a symmetric matrix in $\mathbb{R}^{K \times K}$, whose average we uses as an inverse of SNR. The intermediate feature separation is then characterized by the minimization of this quantity: $\text{CDNV}_{l,k,k'} \to 0, \forall k \neq k'$ and $l \in [L]$. This alternative measurement is faithful to the $\mathcal{NC}_1$ used in the He & Su (2023) but usually more robust and numerically stable as shown in Figure 4.

- **Maximal between-class separation.** At the last layer, the class-mean centers $\overline{\mathbf{h}}_{L,k}$ centered at the global-mean center $\overline{\overline{\boldsymbol{h}}}_L$ are maximally and equally distant, which exhibits an elegant Simplex Equiangular Tight Frame (ETF) structure: given the constant $c \in \mathbb{R}^+$, $\overline{\boldsymbol{H}}_L = \left[ \overline{\boldsymbol{h}}_{L,1} - \overline{\overline{\boldsymbol{h}}}_L \quad \cdots \quad \overline{\boldsymbol{h}}_{L,K} - \overline{\overline{\boldsymbol{h}}}_L \right]$ satisfies

$$\mathcal{NC}_2 = \left\| \frac{\overline{\boldsymbol{H}}_L^T \overline{\boldsymbol{H}}_L}{\left\| \overline{\boldsymbol{H}}_L^T \overline{\boldsymbol{H}}_L \right\|_F} - \frac{1}{\sqrt{K-1}} \left( \boldsymbol{I}_K - \frac{1}{K} \mathbf{1}_K \mathbf{1}_K^T \right) \right\|_F \to 0. \tag{8}$$

## 4 RESULTS

In this section, we present and analyze the empirical results regarding intermediate neural collapse and its relationship with generalization, robustness, and adaptability. First, we investigate the intermediate neural collapse and its correlation with generalization in Section 4.1. Next, we examine

the progressive neural collapse under noisy data conditions to assess its robustness in Section 4.2. Finally, we explore the relationship between the progressive neural collapse and model adaptability in Section 4.3. Additional experimental details are provided in the Appendix.

## 4.1 PROGRESSIVE NEURAL COLLAPSE AND GENERALIZATION

To investigate intermediate neural collapse, we examine two widely-used architectures: ResNet He et al. (2016) and Swin-Transformer Liu et al. (2021), across four datasets, including CIFAR-10 Krizhevsky et al. (2009), CIFAR-100 Krizhevsky et al. (2009), Mini-ImageNet Vinyals et al. (2016), and ImageNet Deng et al. (2009). For each model, we extract features from the intermediate layers and compute the CDNV and $\mathcal{NC}_2$ metrics to assess within-class variability and between-class separation, respectively. The results for ResNet models are shown in Figure 1, while the results for Swin-Transformer models are presented in Figure 5. These visualizations consistently demonstrate that both within-class variability and between-class separation progressively improve across layers in all architectures. Furthermore, we observe that the rate and pattern of within-class variability vary depending on the network depth and the complexity of the dataset. Within a specific dataset, increasing the number of intermediate blocks leads to a decrease in the within-class variability of individual blocks. Once the model complexity becomes sufficient for the dataset, an interesting pattern emerges: after an initial improvement in within-class variability in the early layers, a plateau is observed in the intermediate layers. Following this plateau, the final layers exhibit a renewed, continuous improvement in within-class variability.

By examining this phenomenon in relation to generalization performance, we perform linear probing on top of each intermediate block. Our findings indicate that increasing model depth yields only marginal improvements in generalization once the plateau phase is reached (e.g., the accuracies of ResNet50, ResNet101, and ResNet152 are 95.55%, 95.58%, and 95.58% on CIFAR-10, and 75.91%, 75.93%, and 76.10% on CIFAR-100, respectively). Unlike conventional approaches that require a separate validation set to determine the optimal depth for maximizing generalization, our results suggest that this transitional behavior can serve as an intrinsic indicator for identifying the most efficient model depth, beyond which further increases provide diminishing returns.

On the other hand, the plateau region followed by accelerated decay of the collapsing measure in the final layers provides a clear characterization of the general belief that the earlier layers focus on learning universal and meaningful features, while the last few layers tend to capture more task-specific features. In the next subsection, we examine the robustness of progressive neural collapse when trained on noisy data.

## 4.2 ROBUSTNESS OF PROGRESSIVE NEURAL COLLAPSE WITH NOISY DATA

Since the seminal work Zhang et al. (2021), which demonstrate that DNNs can memorize random labels, the performance of DNNs on noisy labels has been leveraged to understand their generalization and memorization properties Arora et al. (2018); Feldman & Zhang (2020); Anagnostidis et al. (2022); Song et al. (2022). However, most of this work focuses on the entire network's performance without studying the internal representations. In this work, we examine the internal representations of DNNs trained on noisy data and utilize this analysis to gain insights into generalization and memorization. We will study two noisy settings: label noise and corrupted inputs.

For DNNs trained on a noisy dataset (with either noisy labels or noisy inputs), we evaluate their internal representation learning abilities on both noisy and clean data, specifically using $\mathcal{CDNV}_{\text{clean},l}$ and $\mathcal{CDNV}_{\text{noise},l}$ that represent the $l$-th layer $\mathcal{CDNV}$ computed on the clean and noisy datasets, respectively. We now introduce the notion of *memorization ratio* based on $\mathcal{CDNV}_{\text{clean},l}$ and $\mathcal{CDNV}_{\text{noise},l}$.

**Definition 1 (Memorization ratio)** *For a DNN trained on noisy data, we call the ratio $\Delta_{\mathcal{CDNV},l} = \frac{\mathcal{CDNV}_{clean,l}}{\mathcal{CDNV}_{noise,l}}$ as the memorization ratio of the network at the $l$-th layer.*

Intuitively speaking, when the feature mapping overfits the noisy dataset, $\mathcal{CDNV}_{\text{noise},l}$ becomes small while $\mathcal{CDNV}_{\text{clean},l}$ remains large, resulting in a high memorization ratio $\Delta_{\mathcal{CDNV},l}$. Conversely, if the feature mapping encodes meaningful features, the memorization ratio $\Delta_{\mathcal{CDNV},l}$ will be small. Based on this discussion, we can now define *memorization layers* as follows.

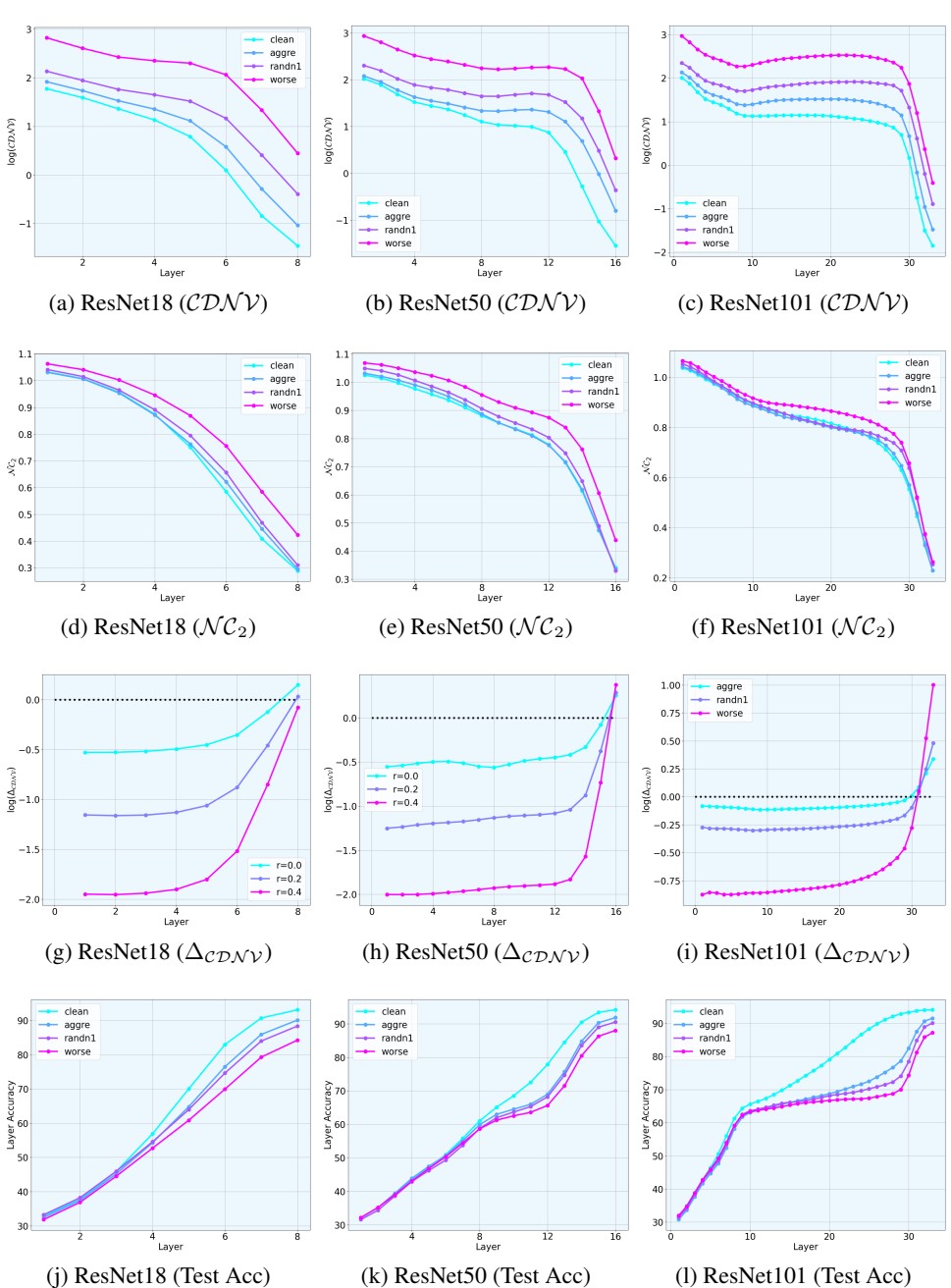

Figure 2: **The evolution of intermediate feature separation across layers for ResNet based models on CIFAR-10N dataset.** The graphs depict the layer-wise progression of within-class variability (top row), between-class separation (second row) using noisy label, memorization ratio $\Delta_{\mathcal{CDNV}}$ (third row) and layerwise linear-probing accuracy (bottom row). The percentage of noisy labels increases in the order: clean, aggre, randn1, worse.

**Definition 2 (Memorization layers)** *For a DNN trained on noisy data, we define the memorization layers as $\{l : \Delta_{\mathcal{CDNV},l} > 1\}$, which often occur consecutively and primarily in the final few layers.*

**Label noise:** We use the real-world, human-annotated CIFAR-10N dataset Wei et al. (2021) and a synthetically generated, randomly labeled CIFAR-10 dataset. We note that our goal is not to propose better training methods or architectures for handling label noise, but rather to gain a deeper understanding of how label noise affects representation learning through standard training. We

visualize intermediate neural collapse across varying noise levels in Figure 2. Additional results are provided in the Appendix. From the figures in the first two rows about feature compression and separation on noisy labels, we observe consistent trends in the improvement of within-class variability and between-class separation across different noise levels, similar to those seen in clean label settings. However, increasing the ratio of noisy data causes the overall curves to shift upward, as the added noise makes it more difficult for the model to separate the intermediate features. This phenomenon suggests that intermediate neural collapse can serve as an effective indicator of the noise level present during model training.

Figure 2(g-i) shows the layer-wise memorization ratio $\Delta_{\mathcal{CDNV},l}$ between the clean and noisy data. We observe that the memorization ratio progressively increases across layers. Notably, regardless of the network size, the memorization ratio remains below 1 for all layers except the final few, indicating that memorization primarily occurs in the last layers. Our results not only align with existing findings that, in partial label noise settings, DNNs demonstrate surprising robustness and generalization performance (Rolnick et al., 2017), but also provide insights from the perspective of internal representations. When comparing two models trained on different noise levels, such as mild label noise ("aggre") and severe noise ("worse") in Figure 2(i), a smaller memorization ratio does not necessarily indicate better performance. Instead, it suggests that the model does not memorize much, even in the presence of higher noise. Recall that the memorization ratio approaches 1 when the noise level is very low.

To illustrate the connection of neural collapse with generalization, we perform linear probing on each intermediate feature using clean data to analyze the impact of noise across different layers. We observe that the gap between models pre-trained with varying degrees of label noise is small in the initial layers but widens as the layers progress deeper, indicating that the initial layers primarily learn features that are less sensitive to noise. As the network progresses, the later layers focus more on task-specific features, making them more susceptible to noise.

**Corrupted input data:** To investigate the impact of corrupted input data, we utilize the CIFAR10-C dataset Hendrycks & Dietterich (2019), which includes various common perturbations. We visualize intermediate neural collapse across different noise levels for Gaussian-type and Speckle-type input perturbations in Figure 8 and Figure 9. Similar to the label noise case, we observe consistent trends in the reduction of within-class variability and the increase in between-class separation across different noise levels, alongside progressive memorization across layers. We also observe slightly more memorization layers, indicating that fitting corrupted input poses greater challenges.

### 4.3 PROGRESSIVE FEATURE SEPARATION AND ADAPTATION

To investigate the correlation between intermediate neural collapse and adaptability in the presence of domain discrepancies between the source and target domains, we evaluate ResNet50 on the Office-Home Venkateswara et al. (2017) dataset, which contains images from four distinct domains: Art, Clipart, Product, and Real-World. Using models pretrained on different source domains, we evaluate the intermediate $\mathcal{NC}$ on the downstream target domain and perform linear probing with a limited amount of downstream data. As shown in the Figure 3, our findings reveal that models pretrained on semantically similar domains display a similar trend of progressive neural collapse. While the differences in the intermediate neural collapse between pretrained models are minimal in the early layers, these discrepancies become more pronounced in the deeper layers. Moreover, we find that the intermediate features have better neural collapse will induces better linear-probing accuracy on the down-stream domain.

## 5 CONCLUSION

In this work, we extend the study of last-layer $\mathcal{NC}$ to the intermediate layers. we conduct an extensive empirical investigation across a diverse set of computer vision datasets, focusing on the intermediate representations of the contemporary neural networks in real-world scenarios. The empirical results reveal that the intermediate layers progressively concentrate features within the same class and separate features between different classes, exhibiting $\mathcal{NC}$ in the final layers and effectively solving the classification task. Moreover, we identify a distinct transition phase in the within-class collapse, where the initial improvement reaches a plateau and additional layers provide minimal

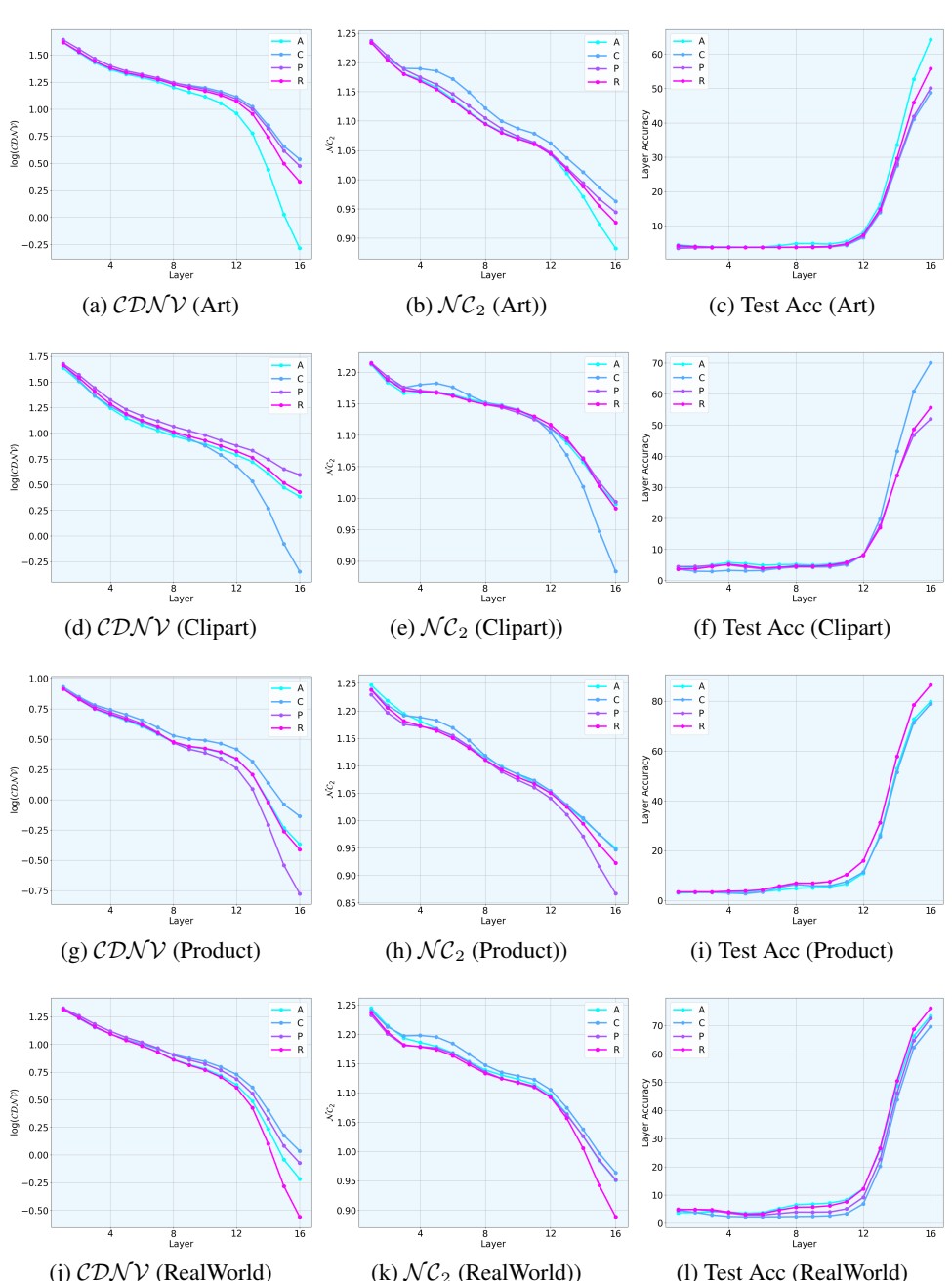

Figure 3: **The evolution of intermediate $\mathcal{NC}$ and linear-probing accuracy across layers for ResNet50 on Office-Home dataset.** Each row plots the intermediate $\mathcal{NC}$ of different target domain.

benefit. Interestingly, after this plateau, the final layers demonstrate a renewed, continuous improvement in within-class collapse. We also observe marginal gains in generalization with the addition of more layers following the transition phase. Additionally, we study the robustness and adaptability of the progressive data compression and separation in the presence of labeling noise, corrupted inputs, and domain shift. For label noise and corrupted inputs, the intermediate features of noisy data still exhibit progressive neural collapse, with patterns remaining similar to those observed in clean data, though the magnitude of neural collapse decreases as the noise level increases. In the case of domain shift, we find that intermediate features exhibiting greater neural collapse on downstream target data tend to demonstrate better adaptability and yield higher linear probing accuracy.

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

# A APPENDIX

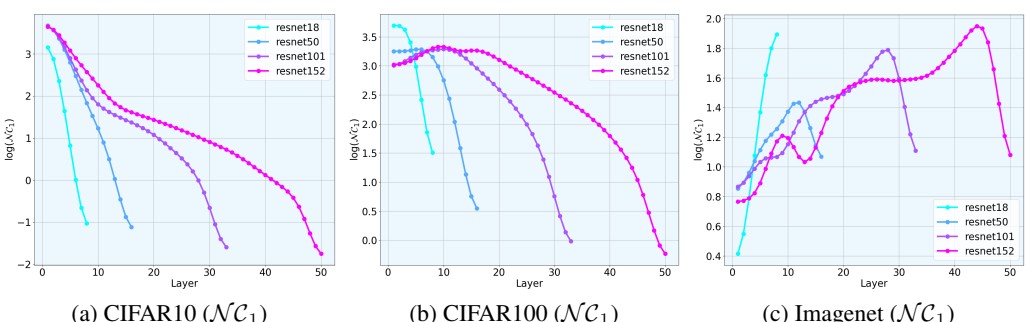

(a) CIFAR10 ($\mathcal{NC}_1$)        (b) CIFAR100 ($\mathcal{NC}_1$)        (c) Imagenet ($\mathcal{NC}_1$)

Figure 4: **The evolution of intermediate $\mathcal{NC}_1$ across layers for different ResNet based models on various datasets.**

**Comparison between $\mathcal{NC}_1$ and $\mathcal{CDNV}$.** The $\mathcal{NC}1$ metric, first introduced in Papyan et al. (2020), has been widely used in subsequent studies on neural collapse. It is defined as trace($\Sigma\mathbf{W}\Sigma_{\mathbf{B}}^{\dagger}$), where $\Sigma_{\mathbf{W}}$ and $\Sigma_{\mathbf{B}}^{\dagger}$ represent the intra-class covariance matrix and the pseudo-inverse of the inter-class covariance matrix, respectively. While both $\mathcal{NC}_1$ and $\mathcal{CDNV}$ measure the ratio of intra-class variability to inter-class separation, we find that the original $\mathcal{NC}_1$ is less stable than $\mathcal{CDNV}$. As shown in Figure 4, although $\mathcal{NC}_1$ demonstrates a progressive reduction in variability on the CIFAR-10 dataset, its pattern varies across more complex datasets and architectures. Therefore, we use $\mathcal{CDNV}$ as an alternative measure of within-class variability.

**More experiments results of progressive neural collapse and generalization.** In Figure 5, we present the intermediate neural collapse on Swin-Transformer based model across CIFAR-10, CIFAR-100 and Mini-ImageNet datasets. From the figures, we can observe that different Swin-Transformer models also consistently enhances the data compression and separation across different blocks. Since the Swin-Transformer was originally designed for large-scale datasets like ImageNet, utilizing various optimization techniques, its optimization on small-scale datasets remains underexplored. As the model continuously improves the intermediate neural collapse without a noticeable phase transition, the performance steadily increases. For example, the accuracies of Tiny, Small, Base, and Large models are 87.17%, 87.58%, and 87.82% and 88.38% on CIFAR-10; and 69.47%, 69.77%, 70.14% and 70.45% on CIFAR-100; and 67.47%, 67.77%, 68.22% and 68.53% on Mini-ImageNet, respectively.

**More experiments results of progressive neural collapse and robustness.** We visualize intermediate neural collapse of ResNet models on CIFAR-10 dataset with ramdom label in Figure 6 and Swin-Transformer models on CIFAR-10N datasets in Figure 7 across varying noise levels. Moreover, we plot the intermediate neural collapse of ResNet models on CIFAR10-C dataset with speckle type input corruption in Figure 9.

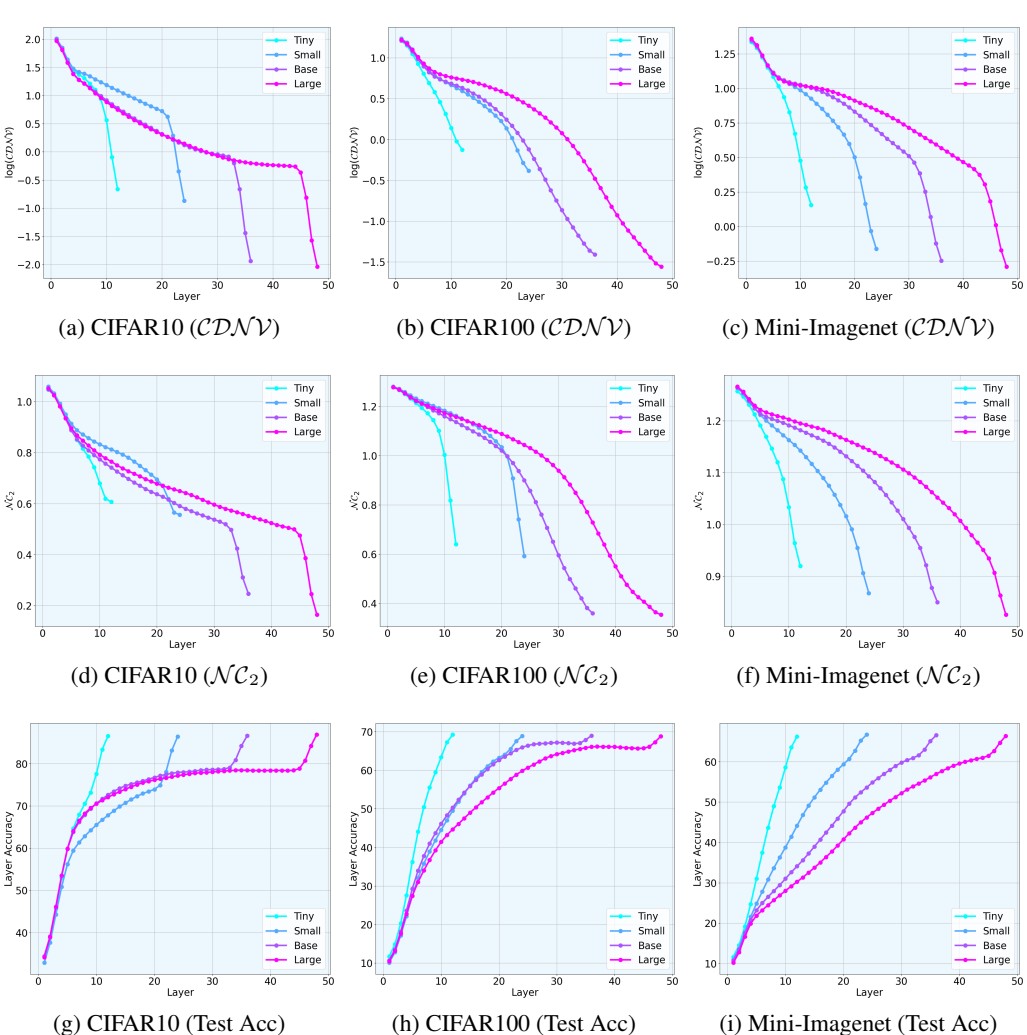

Figure 5: **The evolution of intermediate $\mathcal{NC}$ for different Swin-Transformer based models.**

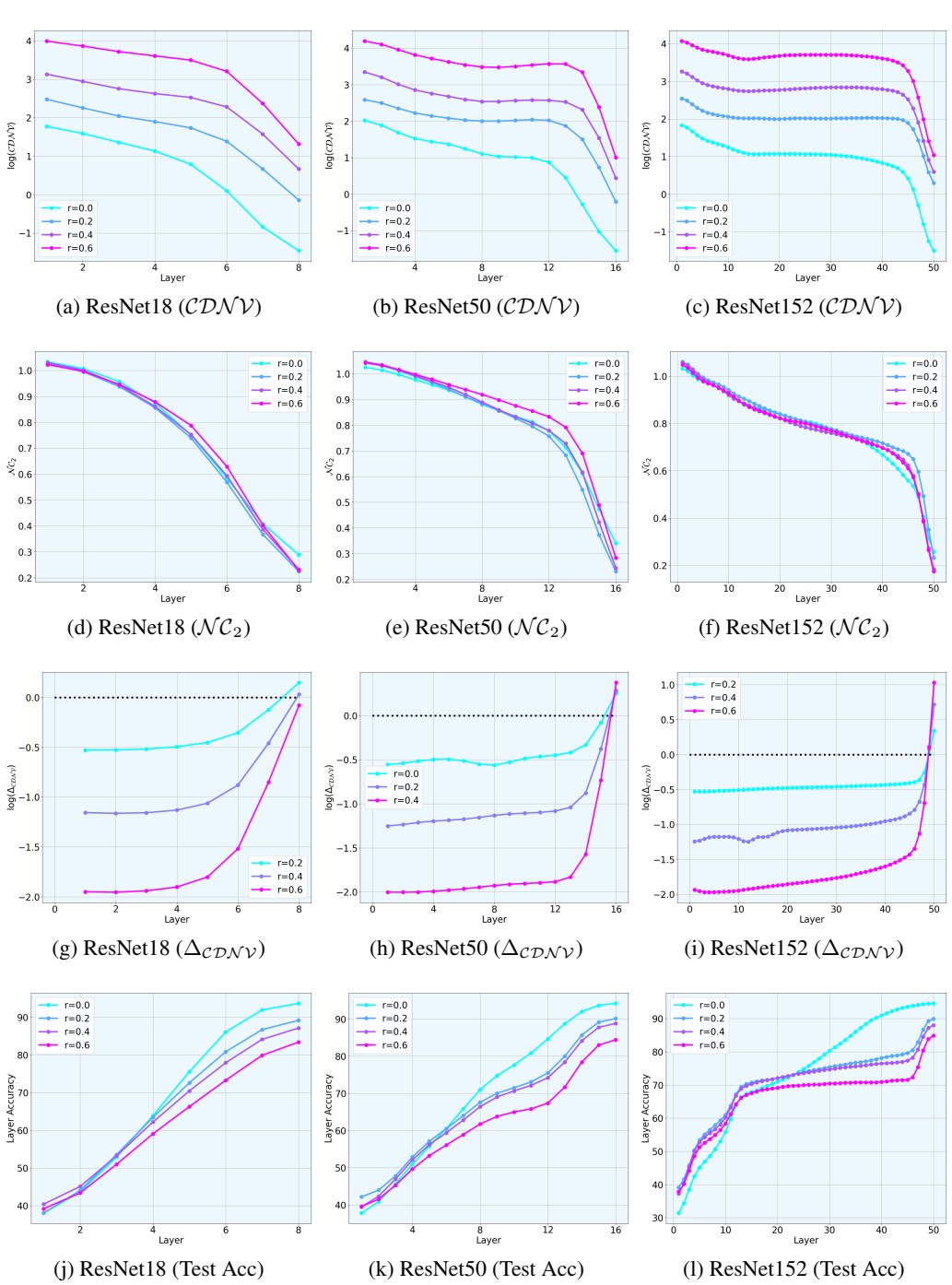

Figure 6: **The evolution of intermediate feature separation across layers for ResNet based models on CIFAR-10 dataset with random labels.** The graphs depict the layer-wise progression of within-class variability (top row), between-class separation (second row) using noisy label, memorization ratio $\Delta_{\mathcal{CDNV}}$ (third row) and layerwise linear-probing accuracy (bottom row) on CIFAR-10 dataset with random noisy labels for different ResNet architectures. We use $r$ to represents the percentage of random labelled data.

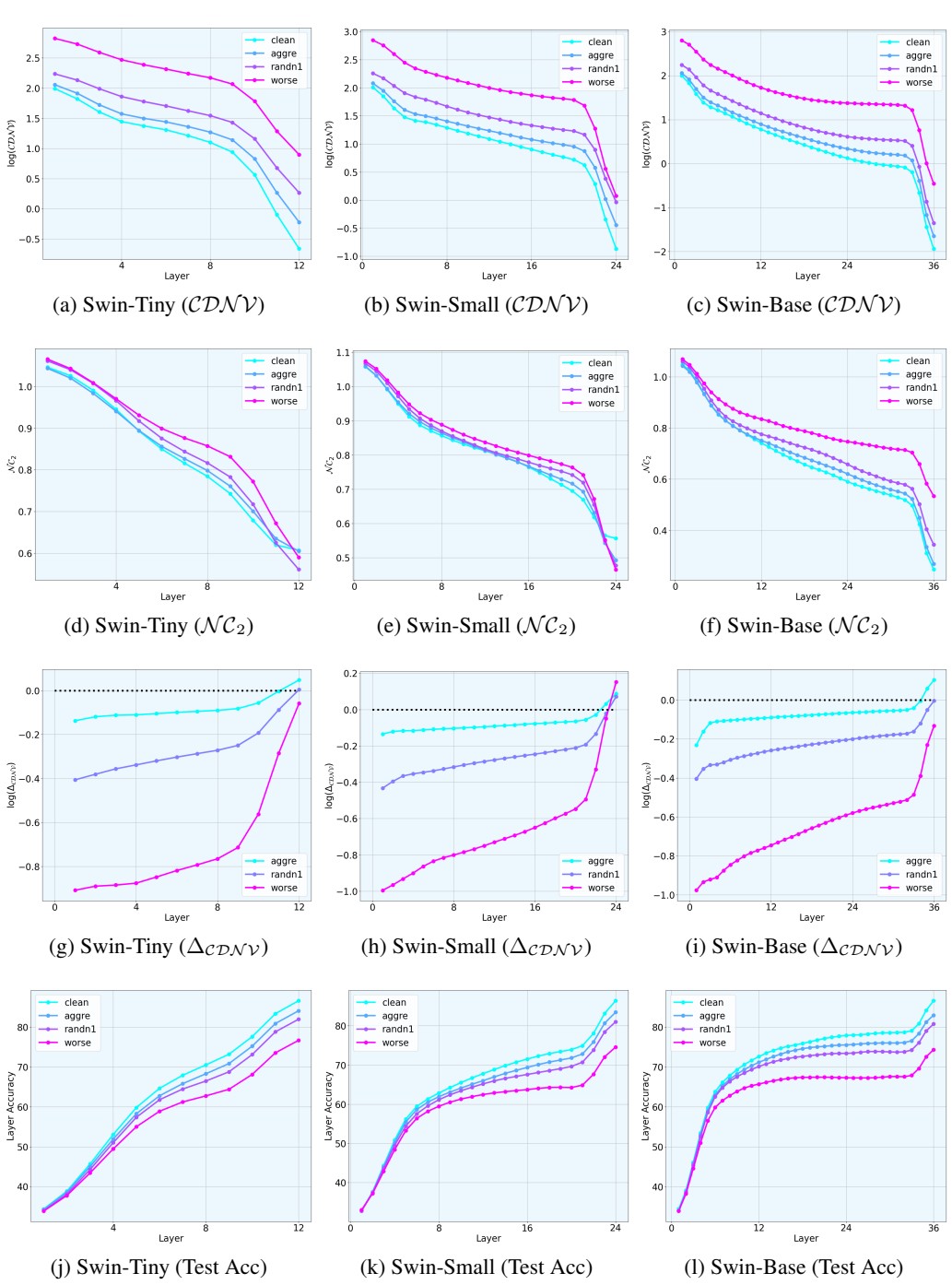

Figure 7: **The evolution of intermediate feature separation across layers for Swin-Transformer based models on CIFAR-10N dataset.** The graphs depict the layer-wise progression of within-class variability (top row), between-class separation (second row) using noisy label, memorization ratio $\Delta_{\mathcal{CDNV}}$ (third row) and layerwise linear-probing accuracy (bottom row) on CIFAR-10N dataset. The percentage of noisy labels increases in the order: clean, aggre, randn1, worse.

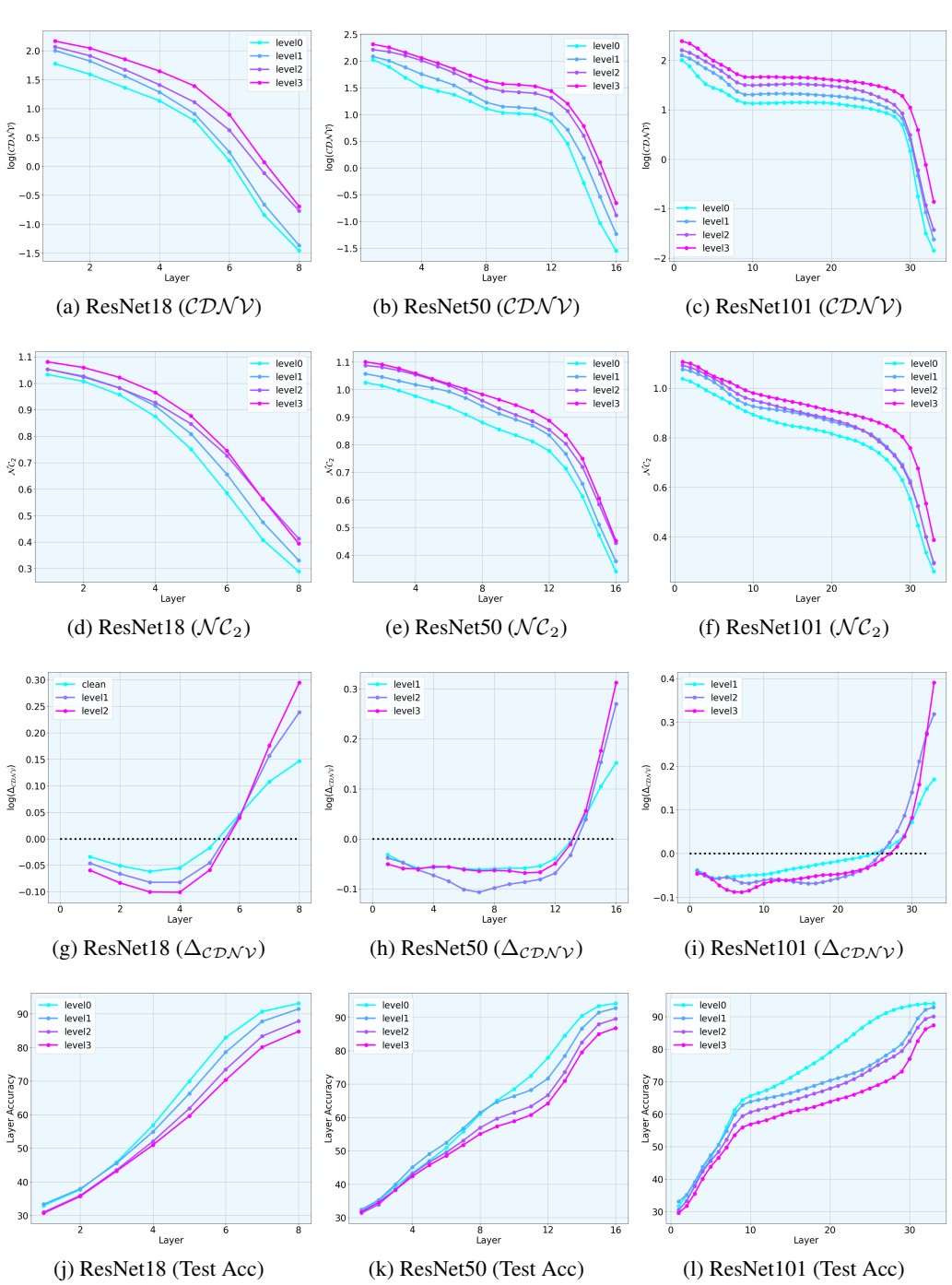

Figure 8: **The evolution of intermediate feature separation across layers for ResNet based models on CIFAR-10C (Gaussian) dataset.** The graphs depict the layer-wise progression of within-class variability (top row), between-class separation (middle row), memorization ratio $\Delta_{\mathcal{CDNV}}$ (third row) and layerwise linear-probing accuracy (bottom row) on CIFAR-10C Hendrycks & Dietterich (2019) (Gaussian noise) dataset for different ResNet architectures. The degree of perturbations increases in the order: level0 → level3.

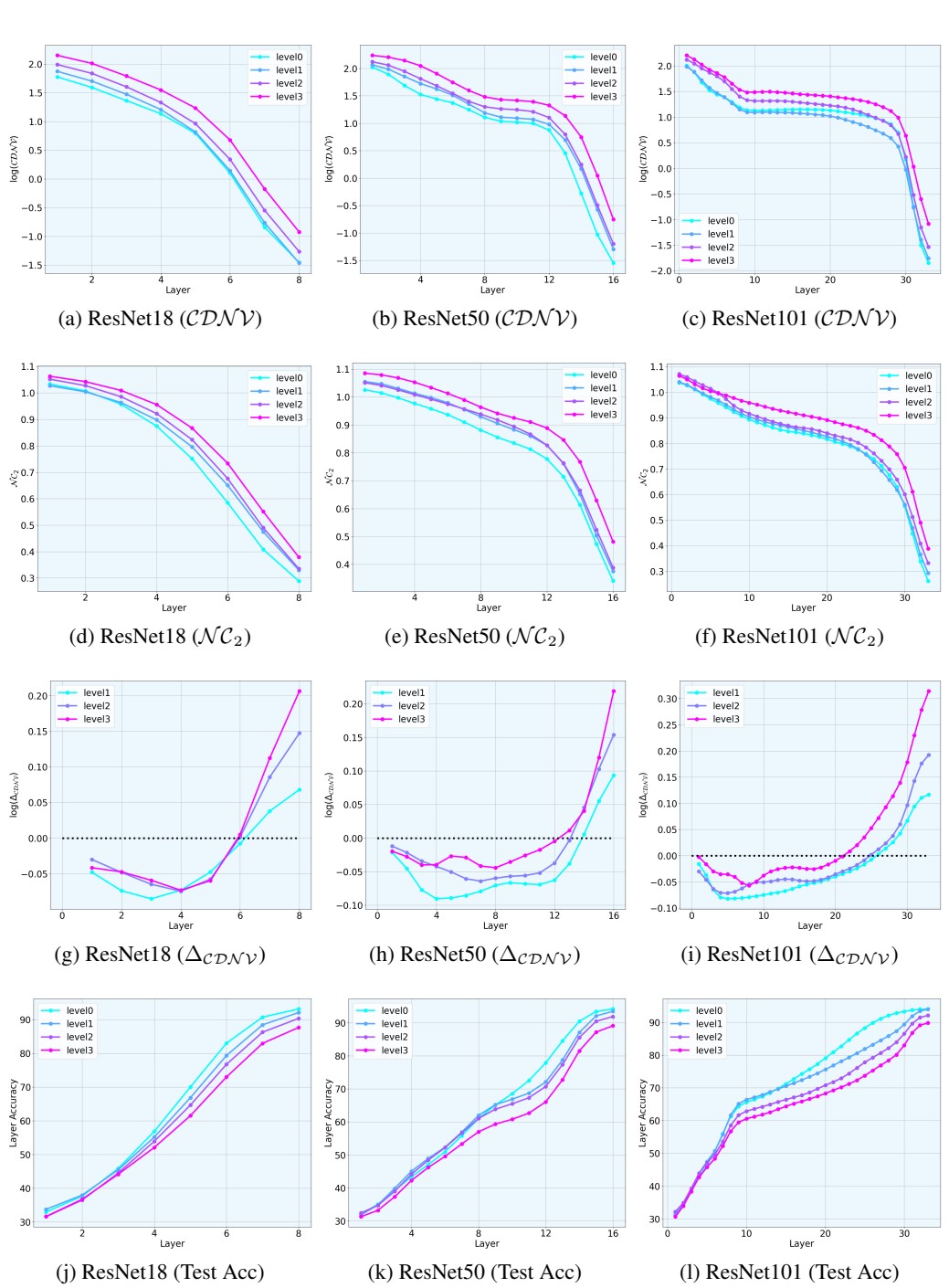

Figure 9: **The evolution of intermediate feature separation across layers for ResNet based models on CIFAR-10C (Speckle) dataset.** The graphs depict the layer-wise progression of within-class variability (top row), between-class separation (middle row), memorization ratio $\Delta_{\mathcal{CDNV}}$ (third row) and layerwise linear-probing accuracy (bottom row) on CIFAR-10C Hendrycks & Dietterich (2019) (Speckle noise) dataset for different ResNet architectures. The degree of perturbations increases in the order: level0 $\rightarrow$ level3.

