# OpenReview forum: "GENERALIZATION, ROBUSTNESS AND ADAPTABILITY OF PROGRESSIVE NEURAL COLLAPSE"
_ICLR.cc/2025/Conference — ICLR 2025 Conference Withdrawn Submission_

### Official Review · Reviewer_Rtzt · 2024-11-01

**Soundness:** 3
**Presentation:** 1
**Contribution:** 1
**Rating:** 3
**Confidence:** 4

**Summary:**

The paper aims to study deeper the neural collapse, particularly taking into account more layers of the network.
This motivation is, however, not substantially novel.  "Neural collapse Neural Collapse in the Intermediate Hidden Layers of Classification Neural Networks" and other papers have already touched in the subject of analyzing intermediate layers and I do not see how the findings here improve our understanding.

Moreover, the paper suffer from a poor presentation, with only two result figures. It seems it is more interested in results than diving deeper into the reasoning and explaining it to a wider audience. Unfortunately I cannot recommend it further.

**Strengths:**

See above.

**Weaknesses:**

See above.

**Questions:**

In which way the current paper progresses the understanding in the field? Use recent papers like  "Neural collapse Neural Collapse in the Intermediate Hidden Layers of Classification Neural Networks" to justify your argument.
And how novel is this given that main motivation of studying neural collapse beyond the last layers have already been explored?

---

### Official Review · Reviewer_AWk9 · 2024-11-03

**Soundness:** 3
**Presentation:** 2
**Contribution:** 2
**Rating:** 5
**Confidence:** 3

**Summary:**

The authors present an empirical study of neural collapse as a function of model layer across deeper models than previously studied.
The authors show that a neural collapse occurs in two phase transitions in larger models - at early layers and at next-to-last layers.
The authors also show relation between the values of NC properties (within-class variability, inter-class separation) of the activations in each layer of the model to several measurements (generalization to test data, memorization of noisy samples, adaptation to different input data distribution).

**Strengths:**

The authors present an empirical study into neural collpase as a function of model layer, which is different than the common approach to investigating NC, which is across training steps. This is an interesting angle, and applying it to models larger than those usually investigated, in real settings, can lead to interesting understandings about real-applied models and not just in the theoretical domain.

The writing is generally good - enough background is given, and the aim to research this specific angle of NC is grounded in related work. Deriving conclusions between the connection of NC to phenomena such as generalization and memorization in classification models is also a novel angle (to the best of my knowledge).

**Weaknesses:**

Generally, the paper requires further refinement. This is exemplified in the following:

1. While the authors present interesting empirical conclusions, they seem rather detached from other existing works in the fields of generalization and memorization, and it would be nice to see some discussion regarding it. For example, the authors claim to use NC measures to claim that memorization occurs in later layers, which is contradictory to some findings, such as [1], that localized memorization across all model layers. This makes me wonder if the measurement given for "memorization" by the authors is faithful enough to support their claims.

2. Some of the experimental settings and results aren't clear. For example -
    1. Do the authors train the base models themselves from scratch? I assume so, but it would be nice to explicitly mention this.
    2. Why are some experiments utilizing ImageNet while others utilize Mini-Imagenet? Which classes and data sample amounts are used in the ImageNet experimentation?
    3. In fig2, the exact amount of noise in each setting "clean, aggre, randn1, worse" is missing. The authors should specify exatcly the percentages of noise in each setting.

3. Writing: while the writing is generally understandable, it can be made more succinct and much clearer. Many parts feel over-convoluted, and can be written in much simpler terms. For example, the entire second contribution paragraph could be summarized to a single sentence. This over-convolution also leads to the fact that the authors' new contributions only appear towards the end of page 6.
I believe a different order of presentation would make the paper more clear and allow it to be more concise - namely, section 3.3, where neural collapse is properly explained, should come earlier (before the discussion of related works, which assume familiarity with different aspects of neural collapse). This can help in removing repetitions on the same subject.
Another point that was a bit unclear due to over-convoluted writing was the term "progressive". Since NC is usually observed in terms of training steps, "progressive" initially hints at training procedure, while the authors use it for "deeper in the network". This is a bit confusing in a first read. Also, the terming of "improvement" for NC values is confusing - each term should be stated as going higher / lower instead of "improving".

Additionally, much proofing is required. Some minor examples:
	1. In fig1: "dataset" -> "datasets" (in bold, line 88). Various spaces are missing.
	2. Line 202: "h_{L+1}" should be "b_{L+1}"
	3. Line 213-215: "entire networks" -> "entire network", "the entire parameters" -> "all parameters", "the all parameters" -> "all the parameters", "the network parameters" is written twice.
	4. In Line 270, I assume the authors meant (P^S != P^T and not P^T != P^T).
	5. Line 474: "features have" -> "features that have", "induces" -> "induce".
        6. The appendix should be split to several sections, each on a specific topic. When pointing to it, the authors should point to a specific figure / section, and not (like in line 432) point to the appendix generally.


[1] https://arxiv.org/abs/2307.09542

**Questions:**

1. In line 225-226, the authors write "when acquiring label for the target label is difficult ".  Do the authors refer to cases where the ground truth label doesn't exist? or something else?
2. In the second contribution, the authors say that adding more layers after the "transition layer" only leads to marginal gains. But in the third, they argue that the existence of the plateau in middle layers benefits generalization of DNNs. Isn't this contradictive?
3. In figure 5, it seems the authors conclusion regarding 2-step phase transition doesn't hold for SWIN-Transformers for CIFAR100. To my understanding, CIFAR100 is considered a less complex dataset than Mini-Imagenet, which does show the 2-step phase transition. Since this comes as a contradiction to one of the main conclusions, I expected to see some discussion regarding this. Do the authors have an explanation?

---

### Official Review · Reviewer_vmNK · 2024-11-03

**Soundness:** 2
**Presentation:** 3
**Contribution:** 2
**Rating:** 5
**Confidence:** 4

**Summary:**

The paper explores the phenomenon of neural collapse of representations across various layers of deep networks trained on vision datasets. The paper empirically studies how features collapse to class means in a progressive manner across layers, and finds that the collapse exhibits a phase transition, plateauing for middle layers before rapidly increasing for the last few layers. The paper correlates this with test accuracy of linear probes across layers, and across model sizes, claiming that for datasets where inter-class variability plateaus, adding more layers does not lead to much better test performance. Finally, the paper empirically looks at progressive neural collapse under domain shifts.

**Strengths:**

1. Findings and direction of analysis of intermediate features is novel and interesting. In particular, explaining some trends of OOD robustness using this framework is also actionable.
2. The paper presents a nice overview of existing literature in NC.

**Weaknesses:**

1. The exact relationship between “plateau” and performance saturation seems vague. Looking at Swin-T on CIFAR 100, no plateau is seen, as opposed to to ResNets on CIFAR-100. In both cases, performance improves for larger models, but the paper claims that it saturates for ResNets, but not for Swin-Transformers. Some correlation metric between a definition of plateau (e.g. based on consecutive values of CDNV) and accuracy across model sizes could quantify this.
2. The insight described above is not also very actionable, since the plateauing happens mainly for the larger models. This means that in order to figure out whether large models would saturate in performance, one needs to train a large model anyway.

2.1 There also isn’t a lot of intuition in the paper as to why only larger models show this plateau phenomenon, and any hypotheses on this would strengthen the paper.

3. Line 474-475 seem unsubstantiated by evidence, claiming “intermediate features [that] have better neural collapse will induce better linear probing accuracy”. I cannot directly see why this is true based on Fig 3.

4. Organization can be made better.

4.1 Background on deep learning is not needed for this audience.

4.2 Transformers results are pushed to the appendix, despite the paper dubbing it to be a major contribution.

4.3 Nit - Figures are far away from the text describing them. Also, adding the takeaway in the caption will make it more reader friendly.


5. Definitions are non-standard, which makes it slightly laborious to follow the paper. This is not a deal breaker however.

5.1 Domain adaptation settings have names in prior work - label shift, covariate shift etc. The domain adaptation considered in this paper is specifically “supervised domain adaptation”, which the paper should mention.

5.2 The definition of ,emorization is pretty non-standard as compared to previous work

5.3 Nit - memorization ratio should be separately considered for input and output corruption

**Questions:**

1. Do the authors have any intuition as to why the correlation between NC plateau and performance saturation exists in the first place.
2. Could the authors quantify this correlation somehow to make it more rigourous and actionable?
3. Could the authors explain “why intermediate features [that] have better neural collapse will induce better linear probing accuracy”?

---

### Official Review · Reviewer_4uXG · 2024-11-03

**Soundness:** 2
**Presentation:** 2
**Contribution:** 2
**Rating:** 3
**Confidence:** 3

**Summary:**

The authors study the problem of Neural Collapse (NC) in intermediate layers under more realistic and complex settings than previous works. This includes surveying the four maxims of NC in larger models (up to ResNet151, Swin) and datasets beyond CIFAR-10. The authors challenge the notion of progressive NC propagating at a constant geometric rate throughout the network layers, particularly when networks are more complex/deep than ResNet18. They observe a phase transition as depth increases and further investigate its effects on generalization, as well as the presence of this pattern under noisy data conditions.

**Strengths:**

- The basic premise of validating NC on a larger scale is well-motivated.
- Connecting NC with generalization performance appears promising.

**Weaknesses:**

* The work's premise isn’t evaluated thoroughly. The "complex" settings introduced here seem limited to larger models (e.g., moving from ResNet18 to ResNet152). Dataset complexity is also sparsely evaluated; for example, while ResNets are evaluated on ImageNet, Swin isn't. Given the ongoing debate about robustness properties of CNN and transformer-based architectures for vision applications, a more thorough investigation between the two families of models would be appreciated.
* The relationship between NC$_1$ and CDNV isn't precisely clear. For example, results in Figure 4 show that NC$_1$ behaves quite differently on ImageNet compared to CIFAR, while CDNV evaluated on ImageNet exhibits a similar pattern to that on CIFAR datasets. What explains this difference? The authors mention robustness and numerical stability of CDNV, but the discrepancy seems significant: relying on NC$_1$ could lead to different and surprising conclusions compared to relying on CDNV. A more thorough explanation of why this discrepancy exists would be beneficial, as it appears significant.
* While challenging the notion of NC growing at a constant geometric rate is appreciated, the implications of this result are not clear to me.
- Presentation-wise, some parts are confusing: each section (e.g., 4.1, 4.2) presents many points in rapid succession, making it feel more like a collection of facts. Personally, it's hard to gauge their significance.
- The applicability of the approach is difficult to justify. For example, the authors suggest that NC could be used in place of a validation set for model-depth selection, but this does not seem like a practical problem.
- Studying NC under noisy data is interesting. However, the introduced definition of memorization (Definition 1) appears non-trivial, and its rationale is not provided. Interpreting results based on this definition is challenging. The claim that memorization mostly takes place in the last few layers is significant, but I’m not convinced of its reliability due to the definition of memorization. If the authors provided justification for this definition, I would be open to raising my score.

**Questions:**

1. It seems that the rate of NC depends more on the fraction of layer depth over the total number of layers, rather than on the absolute index of the layer. For example, in Figure 1, ResNet18 and ResNet50 appear to evolve similarly as a function of the fraction of layer depth, differing from deeper networks. Have you considered conducting your analysis based on this quantity?
2. Eq. (8): What role does the constant $c$ play?
3. Is depth the only factor of interest? Scaling width is equally intriguing. For instance, if you compared with Wide ResNets, would you observe a similar pattern?

---

### Note · Authors · 2024-11-20

I have read and agree with the venue's withdrawal policy on behalf of myself and my co-authors.